# Polydopamine-Coated Poly-Lactic Acid Aerogels as Scaffolds for Tissue Engineering Applications

**DOI:** 10.3390/molecules27072137

**Published:** 2022-03-25

**Authors:** Ramona Orlacchio, Simona Zuppolini, Iriczalli Cruz-Maya, Stefania Pragliola, Anna Borriello, Vincenzo Guarino, Rosalba Fittipaldi, Mariateresa Lettieri, Vincenzo Venditto

**Affiliations:** 1Department of Chemistry and Biology, INSTM Research Unit, University of Salerno, Via Giovanni Paolo II 132, 84084 Fisciano, SA, Italy; rorlacchio@unisa.it (R.O.); spragliola@unisa.it (S.P.); 2Institute of Polymers, Composites and Biomaterials, National Research Council of Italy, P.le Fermi, 80055 Portici, NA, Italy; simona.zuppolini@cnr.it (S.Z.); cdiriczalli@gmail.com (I.C.-M.); 3CNR—SPIN, Via Giovanni Paolo II 132, 84084 Fisciano, SA, Italy; rosalba.fittipaldi@spin.cnr.it (R.F.); mariateresa.lettieri@cnr.it (M.L.)

**Keywords:** poly-lactic acid, polydopamine, aerogels, scaffolds

## Abstract

Poly-L-lactic acid (PLLA) aerogel-based scaffolds were obtained from physical PLLA gels containing cyclopentanone (CPO) or methyl benzoate (BzOMe) molecules. An innovative single step method of solvent extraction, using supercritical CO_2_, was used to achieve cylindrical monolithic aerogels. The pore distribution and size, analyzed by SEM microscopy, were found to be related to the crystalline forms present in the physical nodes that hold the gels together, the stable α’-form and the metastable co-crystalline ε-form, detected in the PLLA/BzOMe and PLLA/CPO aerogels, respectively. A higher mechanical compressive strength was found for the PLLA/CPO aerogels, which exhibit a more homogenous porosity. In vitro biocompatibility tests also indicated that monolithic PLLA/CPO aerogels exhibited greater cell viability than PLLA/BzOMe aerogels. An improved biocompatibility of PLLA/CPO monolithic aerogels was finally observed by coating the surface of the aerogels with polydopamine (PDA) obtained by the in situ polymerization of dopamine (DA). The synergistic effect of biodegradable polyester (PLLA) and the biomimetic interface (PDA) makes this new 3D porous scaffold, with porosity and mechanical properties that are tunable based on the solvent used in the preparation process, attractive for tissue engineering applications.

## 1. Introduction

In the last two decades, trends in tissue engineering (TE) and regenerative medicine have been addressed to basically overcome the conventional strategies used to replace an organ (i.e., autograft or allograft methods) [1,2], embracing the idea to use biocompatible materials with negligible effects on the immune system response and implant rejection.

Hence, an increasing interest is emerging to conceive three-dimensional structures or scaffolds that are able to support the regeneration of living tissues. Indeed, their peculiar architecture allows for the mimicking of the structural organization of the extracellular matrix, promoting cell colonization and new tissue regrowth [3,4]. In this view, several manufacturing techniques have been recently investigated to tailor 3D architectures with highly controlled pore size distributions. Among them, it is possible to mention supercritical fluid extraction, gas foaming, particulate leaching, thermal-induced phase separation (TIPS), electrospinning, and 3D-printing [5,6,7,8,9,10,11,12,13,14].

Among the pivotal requirements, the scaffold has to be safe for the human body—i.e., not toxic to cells—and it needs to show a degradation rate comparable to the new forming tissue rate, in order to assure a gradual support of mechanical loads during the regeneration/repair process. For this purpose, biodegradable polymers, such as synthetic polyesters, have been preferred due to their tunable decomposition mechanisms through the hydrolysis of the ester linkages [15,16].

In this work, stereoregular poly-L-lactic acid (PLLA), a synthetic thermoplastic polymer approved by the Food and Drug Administration (FDA), was used due to its attractive properties in terms of biodegradability, biocompatibility, and bioresorbability, making it a good choice for various biomedical uses (i.e., bioresorbable sutures or implants, bone fixation devices, and drug delivery systems) [17].

PLLA is a semicrystalline polymer presenting different polymorphic forms depending on the crystallization conditions. The most important are α (and the disordered α’), β, γ and δ [18,19,20,21,22,23]. In recent years, a metastable PLLA crystalline structure, named the ε-form, has been discovered [24].

This form is a co-crystalline structure, in which the host–guest interactions, between host PLLA chains and small guest molecules (commonly organic solvents) included within the crystal lattice, support the structure itself [24,25,26]. The removal of the solvent included in the co-crystals results in the transition to the stable PLLA α-form [24,25,26].

It is worth noting that the high stereoregularity of the PLLA used in this work increases its long-term stability, thus also influencing body re-absorption kinetics and metabolic mechanisms.

On the other hand, it is well known that commercial PLA presents a poor affinity with cells [27]. Therefore, different integration strategies have been proposed. In detail, strategies based on the integration of instructive proteins (i.e., collagen, gelatin, and keratin) into porous materials are considered a valid solution to improve cell–material interactions [28,29].

In this context, polydopamine (PDA) is emerging as a bio-inspired adhesive material able to improve the biorecognition and bioelectrical properties of scaffold surfaces. The PDA coating is extremely easy to deposit with a high adhesion to a wide range of substrates through the self-polymerization of dopamine in an aqueous solution. The chemical structure of PDA—rich in catechol and amine functional groups—mimics that of mussel adhesive proteins, providing this material with a high chemical versatility suitable for further functionalization, high hydrophilicity, and optical and electrical properties [29,30,31,32,33]. Hence, PDA has been largely used as bioconductive interface to increment cell adhesion and modulate cellular responses, as well as in sensing and biosensing applications via anchor biomolecule linkages [34].

In this work, PDA-coated PLLA aerogels have been investigated as a porous scaffold with bio-instructive surfaces. Monolithic aerogels have been obtained by the supercritical CO_2_ extraction of solvents from starting gels. By appropriately choosing the solvent for preparing the starting gel and setting the process conditions, it is possible to balance aerogel mechanical and structural properties at a microscale level. Moreover, the presence of the PDA coating improves the cell interface and the in vitro viability cellular response, with respect to uncoated samples, suggesting their use as cell instructive scaffolds for tissue engineering applications.

## 2. Results and Discussion

### 2.1. PLLA Gels and Aerogels Preparation

Monolithic poly-L-lactic acid (PLLA) aerogels were prepared from PLLA gels containing cyclopentanone (CPO) and methyl benzoate (BzOMe) by a simple one-step solvent extraction procedure using supercritical CO_2_. Aerogel preparation is schematized in Figure 1.

All gel samples were prepared in hermetically sealed test tubes by heating the mixtures at a temperature close to the solvent boiling point until complete polymer dissolution. Once the gel was obtained, the solvent was extracted with supercritical CO_2_. Monolithic cylindrical aerogels were thus obtained with the same dimensions of the starting gels.

### 2.2. WAXD Analysis of PLLA Gels and Aerogels

In Figure 2a, the X-ray diffraction (WAXD) patterns of PLLA gels in CPO (PLLA/CPO, top) and in BzOMe (PLLA/BzOMe, bottom) are shown, respectively. The WAXD pattern of the PLLA/CPO gel presents diffraction peaks at 2θ = 11.1, 14.2, 18, and 22°, which are typical of the PLLA ε-cocrystalline form [25,35]. Therefore, PLLA/CPO gel is held together by physical knots consisting of a clathrate crystalline form in which PLLA is the host phase and the CPO molecules are the guests in the PLLA crystal lattice [36,37,38,39].

The most intense diffraction peaks of the PLLA/BzOMe WAXD pattern are instead located at 2θ = 16.5 and 18.9°, which are typical reflections of the PLLA disordered α’-form [18,22,40]. Therefore, the physical knots of the PLLA/BzOMe gels are made up of PLLA crystallites in the α’ form, which is a crystalline form not including solvent molecules.

It is worth noting that α (the corresponding ordered crystalline form of the α’-form) and ε crystals have both orthorhombic unit cells, but that of the ε-form is larger (a = 1.6 nm, b = 1.26 nm, c = 2.9 nm vs. a = 1.066 nm, b = 0.616 nm, c = 2.888 nm) [24,25].

After solvent extraction with supercritical CO_2_, both the aerogel WAXD patterns, shown in Figure 2b, present the reflections of the PLLA α-form [20], the most intense of which are at 2θ = 16.7 and 19.1°.

These results are in agreement with those reported by Marubayashi et al. [24], who showed that the PLLA ε-crystal form is stable only in the presence of specific solvents, and a solid-solid phase transition, from the ε- to α-form, occurs after the solvent removal. In detail, Marubayashi et al. reported that some organic molecules with a five-membered ring structure (CPO, DOL, GBL, and THF), and a small planar molecule, DMF, containing oxygen, are capable of forming the co-crystalline ε-form [24]. Computational analyses showed that the ε-form has channels delimited by four parallel chains of PLLA [24]. The weak interactions binding these four PLLA chains with eight solvent molecules, embedded into the channels, stabilize the co-crystal structure. More recently, Rizzo et al. demonstrated the existence of channels in the PLLA/cyclopentanone (CPO) co-crystal structure by analyzing CPO-treated PLLA-oriented films [25].

### 2.3. Morphological Analysis

SEM micrographs of PLLA/BzOMe and PLLA/CPO aerogels, obtained from the PLLA/BzOMe and PLLA/CPO gels, respectively, after solvent extraction by supercritical CO_2_, are reported in Figure 3 at different magnifications (1300× and 6000×).

The SEM micrographs of the PLLA/BzOMe and PLLA/CPO aerogels show different morphologies depending on the solvent used to prepare the native gel. In detail, the PLLA/CPO aerogels (Figure 3a,c) have a structure consisting of the homogeneously distributed interconnected nanosheets. On the contrary, the SEM images of the PLLA/BzOMe aerogels (Figure 3b,d) have a non-homogeneous structure consisting of spheroids of about a 30-micron radius, in turn composed of polymer nanosheets.

The homogeneity observed in the PLLA/CPO aerogel is possibly due to the strong interactions present into the starting gel structure between CPO molecules and PLLA lactide units. These interactions, which support the co-crystalline structure and are also partially present in the amorphous phase, probably make the distribution between the amorphous and crystalline phases and, therefore, the microstructure as a whole, more homogenous.

### 2.4. Characterization of Aerogel Porosities

#### 2.4.1. Analysis of Pore Size Distribution by SEM

The aerogel SEM micrographs were analyzed using the data acquisition software SPIP™, in order to determine the pore size distribution in the PLLA/CPO and PLLA/BzOMe aerogels. The software allows the identification of the pores which, based on Iupac nomenclature, have size dimensions of >50 nm (macropores) [41].

The macropore size distribution, evaluated as the percentage of pores (assumed spherical) detected on the analyzed surface for both SEM micrographs of Figure 3c,d, is reported in Figure 4a,b. A clear difference in size distribution for the PLLA/CPO and PLLA/BzOMe aerogels can be seen in the graphs. In detail, a larger distribution of the macropore size is present in the PLLA/BzOMe, while macropores of a small size (< 1 micron) are prevalent in PLLA/CPO. As shown in Figure 4a,b, the most frequent pore size is lower for PLLA/CPO (0.46 µm) than for PLLA/BzOMe (1.14 µm). The cumulative amount of pores vs. pore size (assumed to be spherical) is instead reported in Figure 5. The SPIP^TM^ analysis shows that 98% of the macropores in the PLLA/CPO aerogel have a diameter of less than 1 micron, while in the PLLA/BzOMe aerogel, only 43% of macropores fall into this category.

In conclusion, the SPIP^TM^ software analysis shows a narrower macropore size distribution for the PLLA/CPO aerogel compared to the PLLA/BzOMe one, and substantially supports the hypothesis, suggested by the morphological analysis, that the PLLA/CPO aerogel has a more homogenous structure. Moreover, in the PLLA/BzOMe aerogel, a greater amount of large pores are present, so the macroporosity is higher.

#### 2.4.2. Analysis of Nitrogen Adsorption–Desorption Isotherms

Aerogel porosity was further investigated by analyzing the nitrogen adsorption–desorption isotherms. The surface area and the volume associated with the mesoporosity (pore sizes ranging from 2 to 50 nm) [41] were evaluated with the Brunauer–Emmett–Teller (BET) model, while the Dubinin–Redushkevich (DR) and Barret–Joyned–Halenda (BJH) models were used to obtain the data related to microporosity (pore sizes < 2 nm) [41]. In Table 1, the surface area and pore volume for both micro- and mesopores are reported. It is worth noting that although the volumes of both meso- and micropores are greater in the PLLA/BzOMe than in the PLLA/CPO aerogel, the meso/micropore volume ratio is smaller in PLLA/CPO than in PLLA/BzOMe. In other words, the volume occupied by the meso- and micropores is more comparable in PLLA/CPO than in PLLA/BzOMe; therefore, the volumetric contribution of the meso/micro pores to the global porosity is more homogeneous in PLLA/CPO than in PLLA/BzOMe.

In conclusion, the BET, HD, and BJH analyses of nitrogen adsorption–desorption isotherms show that meso- and micropores occupy more comparable volumes in the PLLA/CPO than in the PLLA/BzOMe aerogels, and suggest that the global distribution of volumes (for the various porosity ranges) is more homogeneous in the PLLA/CPO aerogel than in the PLLA/BzOMe aerogel.

### 2.5. Mechanical Analysis

The compressive strength of the PLLA/CPO and PLLA/BzOMe aerogels was analyzed. The stress–strain curves (for compressive stresses) of the two aerogels are shown in Figure 6. Young’s compression modulus, calculated from the slope of the initial part of the stress–strain curve, appears to be very different in the two samples. In detail, the PLLA/CPO aerogel has an elastic modulus that is almost double compared to the PLLA/BzOMe sample; 5.5 MPa and 2.6 MPa, respectively. Moreover, the toughness of the PLLA/CPO aerogel is greater than the PLLA/BzOMe one, as well as the yield point for the sample PLLA/CPO, which is at higher stress and strain values than those of PLLA/BzOMe.

These results are in good agreement with the indications of the porosimetric analysis, as it is generally accepted that material mechanical properties also depend on the porosity and, in detail, on a homogeneous distribution of pore shapes and sizes [42,43]. The tighter distribution of the macro/meso/micropore sizes and, therefore, the more homogeneous the global porosity of PLLA/CPO is, as suggested by morphological, pore size distribution, and BET, HD, and BJH analyses, gives this material greater stability by reducing the points where a fracture can occur.

It is worth noting that both porosity and mechanical properties play a fundamental role in biomedical devices, as it influences the cell adhesion, and therefore the cell colonization of internal/external material surfaces [44]. Porous materials have demonstrated an improvement in the cell–material interaction due to the high surface area and interconnected structure, which is able to mimic the extracellular matrix [45].

### 2.6. Preliminary In Vitro Viability Tests on Bare Aerogels

The influence of the pore size and distribution of both PLLA/CPO and PLLA/BzOMe aerogels on cell responses was evaluated by in vitro viability tests. Cell adhesion was more than 60%, with respect to the control (TCP) for both tested aerogels (Figure 7a). In detail, a significant increase in cell viability was detected in the case of the PLLA/CPO aerogels after 24 h. Accordingly, a significant increase in cell proliferation was recognized in the case of the PLLA/CPO aerogels (Figure 7b), with respect to the PLLA/BzOMe ones, where an arrest of the metabolic activity of cells occurred after 14 days. On the basis of these preliminary evaluations, the PLLA/CPO aerogels have been selected for the optimization of the PDA surface treatment.

### 2.7. PDA Coated Aerogels: Surface Morphology and In Vitro Validation

The PDA coating of the PLLA/CPO aerogels was performed, at room temperature, by immersing aerogel slices in a Tris-HCl solution (pH = 8.5) containing a dopamine precursor which self-polymerizes to PDA, due to free oxygen oxidant action and alkaline conditions. It is worth noting that the use of a weakly alkaline aqueous solution in presence of oxygen still represents the most used conditions, although over the years, several experiments variables (i.e., the nature of the buffer and/or oxidant, the DA concentration, the time reaction, and the type of oxidants) have been exploited [46].

Since the PDA anchoring on a surface is concurrent to oxidative DA polymerization, it is important that the substrate to be covered is present in a solution already in the initial phase of the polymerization. The formation of PDA can be easily followed by the evident color change of the solution due to insoluble dark black precipitates (particles/aggregates) in suspension. Therefore, the PDA coating on the PLLA/CPO aerogels was confirmed, after washing the sample with water, by the darkening of the white surface aerogel substrate.

SEM images of PDA-coated and uncoated PLLA/CPO aerogels are compared in Figure 8. The aerogel interconnected nanosheet structure, before coating, is shown in Figure 8a. In the coated sample, Figure 8b, a granular deposit of PDA onto the aerogel nanosheets is clearly detectable.

To evaluate the influence of the PDA coating, the cell adhesion of human mesenchymal stem cells (hMSCs) after 24 h, and their proliferation, were evaluated. Figure 9a shows that the effect of the PDA coating on cell adhesion was about 80% higher than the control (TCP). Figure 9b confirmed this trend, remarking a significant increase in cell proliferation in the presence of PDA coated samples, after just 7 days.

These results are not surprising, considering the biocompatibility of the used proteins ascribable to their peculiar chemical composition and macro-structure [47] that both actively influence the in vitro regulation of basic cellular interactions with the substrate.

It is worth noting that PLLA has been widely studied as an inert biomaterial for different material applications [48]. The peculiar porous structure of PLLA aerogels, just alone, may contribute to the creation of a friendly environment to promote cell attachment and proliferation, as confirmed by reported in vitro results (Figure 7). In this context, the presence of the one-dimensional phases of PDA onto the aerogel porous surface allows the exposure of bioactive signals that can favorably modulate in vitro cellular behavior. This may be ascribable to multiple properties of PDA in terms of hydrophilicity and bioelectrical properties, singularly investigated in other studies on similar substrates [49,50].

Based on these results, this preliminary study could be propaedeutic to the design of more efficient electro-conductive scaffolds for specialized tissues with bioelectrical functionalities (i.e., muscles and nerves). With this perspective, a more accurate tuning of synthesis conditions may offer a concrete opportunity to reach a right balance between bioactive/bioconductive properties for the fabrication of continuous coatings, with higher availabilities, to graft functional groups for the immobilization of bioactive molecules and an efficient transfer of electrical signals to cells.

## 3. Materials and Methods

### 3.1. Chemicals

PLLA (viscosity = 2.0 dL/g, Mw ~ 260,000 and Mn ~ 170,000) was purchased from Sigma Aldrich (Milan, Italy). Cyclopentanone (CPO), methyl benzoate (BzOMe), dopamine hydrochloride (DA), and Tris(hydroxymethyl)methane (Tris-HCl) were also provided by Sigma Aldrich (Milan, Italy).

All chemicals were of analytical grade and used without further purification.

### 3.2. PLLA Gels and Aerogels Preparation

All gel samples were prepared in hermetically sealed test tubes by using the following procedure. Test tubes (containing a glass pipe with a diameter of 5 mm) were placed in an aluminum heating block. A well-defined amount of PLLA (30% *wt*/*wt*) was dissolved in a solvent (70% *wt*/*wt*), in order to have a polymer/solvent weight ratio of 30/70. Test tubes were heated at a temperature close to the solvent’s boiling point (130 and 180 °C for CPO and BzOMe, respectively) until the appearance of a transparent and homogeneous solution had occurred.

After the heating treatment, the gels were quenched at −22 and 0 °C for PLLA/CPO and PLLA/BzOMe gels, respectively, for 24 h.

Once the gel was obtained, the cylindrical gel portion in the glass pipe was removed and the solvent extracted using a SFX 200 supercritical CO_2_ extractor (ISCO Inc.) under the following conditions: T = 40 °C, *p* = 250 bar, with an extraction time of t = 240 and 180 min for PLLA/BzOMe and PLLA/CPO gels, respectively. Aerogel preparation conditions are schematized in Table 2.

Monolithic cylindrical aerogels were thus obtained with the same dimensions as the relative starting gel.

### 3.3. WAXD Analysis of PLLA Gels and Aerogels

X-ray diffraction patterns were recorded by an automatic Bruker D2 diffractometer operating at a step size of 0.033°, a time/step of 0.2 s, and with a nickel-filtered Cu Kα radiation with a wavelength of 1.54 Å. The 2θ range was from 5° to 40°.

The diffraction patterns of gels were obtained by spreading the gel on the sample holder and sealing it with adhesive polyamide (Kapton) tape, in order to avoid solvent evaporation. The polyamide tape has a low X-ray scattering, which does not disturb the scattering of the polymer gels.

Aerogel WAXD patterns were collected using a cylinder-shaped sample slice, with a diameter of 5 mm.

### 3.4. Morphological Analysis

The internal morphology of the monolith aerogels was characterized, by means of a scanning electron microscope (SEM, LEO Evo50, Zeiss, Oberkochen, Germany), using, as usual, monolith fragments obtained by cryo-fracturing. In order to obtain the highest possible surface resolution, high energy electron beams (20 kV) were used to analyze the aerogels, while low energy beams (10 keV) were used to analyze the PDA-coated aerogels. Before imaging, all the specimens were sputter-coated with gold using a VCR high resolution indirect ion-beam sputtering system. The coating procedure was necessary in order to prevent the surface from charging during the measurement and to increase the image resolution.

### 3.5. Aerogel Porosity Characterization

The aerogel porosity was investigated using two different methods, analysis of aerogel SEM micrographs by Scanning Probe Image Processor (SPIP^TM^, version 6.2.6), and a nitrogen adsorption–desorption isotherm analysis. SPIP™ (Image Metrology A/S, Kgs. Lyngby, Denmark) is an advanced software package for processing and analyzing microscopic images at nano- and microscales. Images at a magnification of 5000× were analyzed for each aerogel; the “Particle and Pore analysis” function was used to detect the pores and to measure their size. The threshold segmentation method was applied to quantify the pore percentages, and the watershed method was used to evaluate the distribution of the pores on the basis of their dimensions. Specific surface area and pore size distributions of the PLLA/CPO and PLLA/BzOMe aerogels were also analyzed by nitrogen adsorption–desorption isotherms, using a Nova Quantachrome 4200e instrument. N_2_ adsorption–desorption experiments were carried out at 77.4 K. The resulting data were fitted with adsorption models: the Brunauer–Emmet–Teller (BET), Dubinin–Redushkevich (DR), and Barret–Joyned–Halenda (BJH) models. The microporous and mesoporous surface areas were evaluated using BET and BJH models, respectively. The DR model was used to determine the micropore volume and surface area.

### 3.6. Mechanical Analysis

Aerogel compressive mechanical properties were determined using cylindrical specimens and Instron 5000 testing instrument equipped with a load cell of 0.5 kN. Cylindrical specimens with a diameter of 5.0 mm and a height of 5.0 mm were used. The top and bottom surface of the aerogel specimens were polished to ensure good contact with the compression fixture. A crosshead speed of 1.00 mm/min was used, and data for five specimens were collected to obtain average values of the compressive modulus.

### 3.7. Preparation and Characterization of PDA-Coated Aerogel

Thin PLLA/CPO aerogel slices, obtained by cutting cylindrical monolithic aerogels, were used for the PDA-coatings. The coating was performed by immersing the aerogel in a 10 mM Tris-HCl solution (pH 8.5) containing DA (2 mg/mL) and gently stirring at r.t. for 24 h. As the aerogel slices float in solution, only the surface at the solution interface was coated. The PDA-coated PLLA/CPO samples were finally washed with bi-distilled water.

### 3.8. In Vitro Viability

Biological assays were performed by using human mesenchymal stem cells (hMSCs, SCC034 from Sigma-Aldrich, Milan, Italy), which were used for the in vitro tests. Firstly, hMSCs were cultured in a 75 cm^2^ cell culture flask in the Eagle’s alpha minimum essential medium (α-MEM) supplemented with 10% fetal bovine serum (Sigma-Aldrich, Milan, Italy), an antibiotic solution (streptomycin 100 µg/mL and penicillin 100 U/mL, Sigma-Aldrich, Milan, Italy), and 2 mM of L-glutamine, incubated at 37 °C in a humidified atmosphere with 5% CO_2_ and 95% air. hMSCs from 5–6 passages were used for the performed tests. For the cell adhesion, hMSCs (5 × 10^4^ cells/mL) were seeded into the samples. Cells seeded the tissue culture plate (TCP) were used as the control. After 24 h, cell culture media was removed to eliminate unattached cells, and then 100 µL of fresh media with 10 µL of the Cell Counting Kit-8 Reagent (CCK-8; Dojindo Laboratories, Rockville, MD, USA) was added per well. After 4 h of incubation, the supernatant was collected to measure the absorbance at 450 nm in a microplate reader. Results are presented as the percentage of adhesion with respect to the TCP.

Then, hMSCs (1 × 10^4^ cells/mL) were seeded to perform proliferation assays using the Cell Proliferation Kit II (XTT, Roche Applied Science, Penzberg, Germany) according to the manufacturer’s instructions at 1, 3, 7, and 14 days. Briefly, at each time point, the cell culture media was removed and changed by 100 µL of fresh medium with 50 µL of an XTT working solution per well and incubated for 4 h in standard conditions. The supernatant was collected, and absorbance measured at 450 nm using a microplate reader. Biological assays were conducted in triplicate. Results are presented as the mean ± standard deviations. Multiple comparisons between the groups were conducted with Tukey’s post-hoc test with an analysis of variance (ANOVA). A value of *p* < 0.05 was considered statistically significant.

## 4. Conclusions

In this work, a versatile and green technological approach, based on the use of supercritical CO_2_, was proposed for the preparation of PLLA monolithic aerogels with tailored porosities and mechanical properties in agreement with previous studies of the group. This technology exploits PLLA peculiar features to form physical gels in which the crystallites of the α- and ε-forms of PLA are gel physical knots. The ε-crystalline form, a clathrate form in which solvent molecules are guests in the crystal lattice formed by host polymer chains, have been shown to have useful properties. In detail, aerogels obtained by extracting CPO from corresponding PLLA/CPO gels with a ε-clathrate crystalline structure exhibit a homogeneous architecture resulting from the assembly of nanosheets and pores. Monolithic PLLA/CPO aerogels exhibit a higher compressive strength and a better cell viability than PLLA/BzOMe ones, obtained from gels in which PLLA is in the α-crystal form (a non-clathrate crystalline form). Furthermore, it has been demonstrated that the functionalization of the aerogel porous surfaces, via PDA one-dimensional phases, allows for the improvement of the bio-interface with cells, as a consequence of the hydrophilic properties of the protein.

From this perspective, PDA synthesis conditions will be properly set to form a continuous coating to impart bioelectrical properties to the aerogel porous surfaces. This approach will be successfully used to promote a more efficient transfer of electrical signals among cells during the in vitro regeneration of highly specialized tissues, such as muscles and nerves.

## Figures and Tables

**Figure 1 molecules-27-02137-f001:**
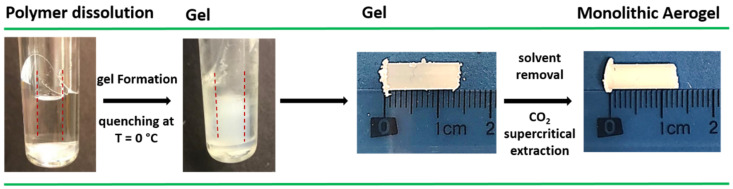
PLLA monolithic aerogel preparation procedure. PLLA dissolution temperature: 130 and 180 °C, with CPO and BzOMe solvent, respectively. Solvent extraction conditions: T = 40 °C and *p* = 250 bar, extraction time = 180 and 240 min, for PLLA/CPO and PLLA/BzOMe gel, respectively.

**Figure 2 molecules-27-02137-f002:**
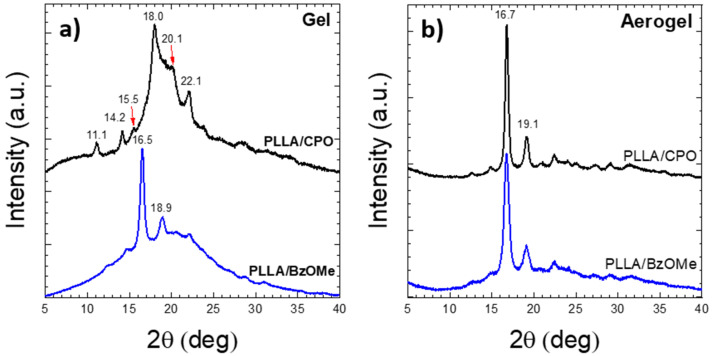
X-ray diffraction patterns of the: (**a**) PLLA gel samples containing CPO (top), or BzOMe (bottom), and (**b**) the relative aerogel samples obtained by solvent extraction with supercritical CO_2_.

**Figure 3 molecules-27-02137-f003:**
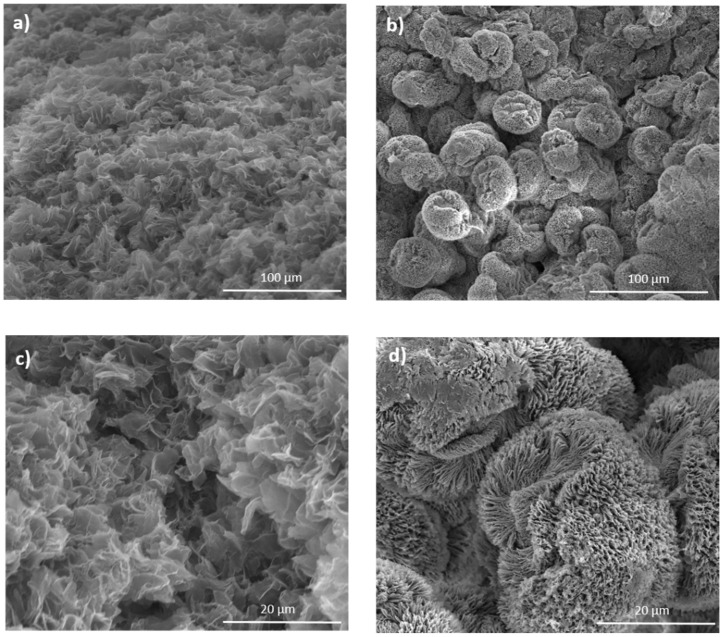
SEM micrographs of aerogels at different magnifications (**a**,**b** = 1300×; **c,d** = 6000×) obtained from gels prepared in CPO (**a**,**c**) and BzOMe (**b**,**d**), at *C_pol_* = 30% *wt*/*wt*.

**Figure 4 molecules-27-02137-f004:**
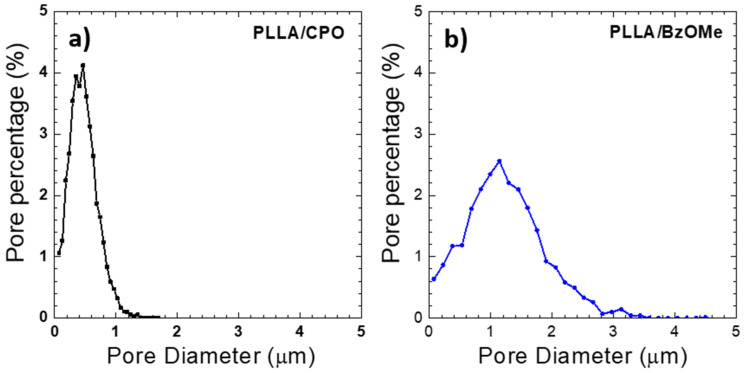
SPIP™ image analysis of SEM micrographs in Figure 3c,d: percentage of pores (assumed spherical) as a function of diameter: (**a**) PLLA/CPO aerogel and (**b**) PLLA/BzOMe aerogel.

**Figure 5 molecules-27-02137-f005:**
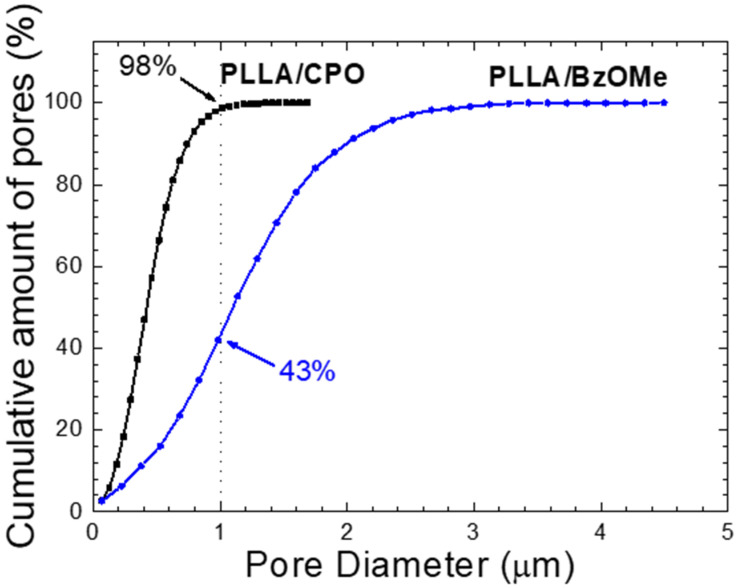
Cumulative amount of pores as a function of pore sizes, obtained by SPIP™ analysis of SEM micrographs reported in Figure 3.

**Figure 6 molecules-27-02137-f006:**
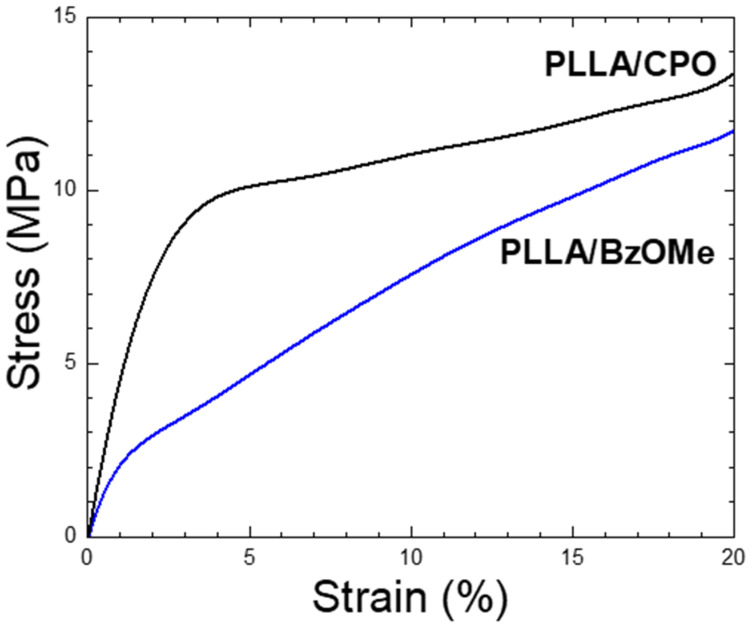
Stress–strain curves by compressive tests of PLLA/CPO and PLLA/BzOMe porous scaffolds, with different porosity and pore morphologies. Each curve is the average of five specimens.

**Figure 7 molecules-27-02137-f007:**
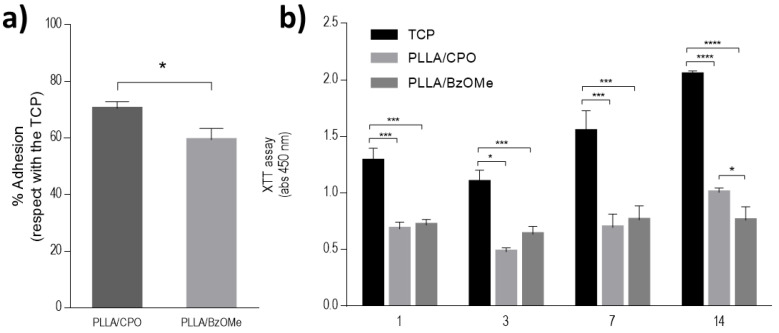
In vitro evaluation of PLLA/CPO and PLLA/BzOMe aerogels. (**a**) Cell adhesion after 24 h. Results show the percentage of adhesion with respect to the control (TCP). (**b**) Cell proliferation after 1, 3, 7, 14 days. (* *p* < 0.05; *** *p* < 0.01; **** *p* < 0.0001).

**Figure 8 molecules-27-02137-f008:**
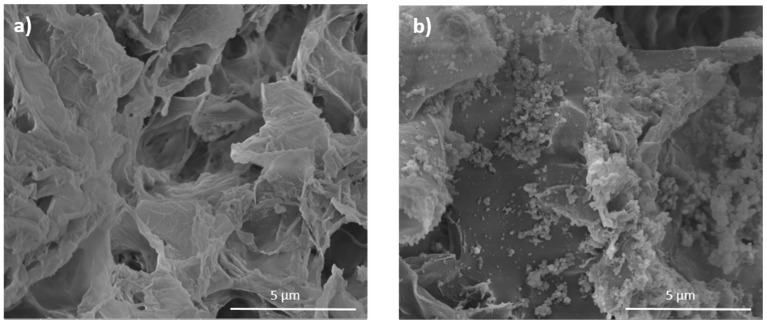
SEM micrographs of PLLA/CPO aerogels without (**a**) and with (**b**) PDA coating.

**Figure 9 molecules-27-02137-f009:**
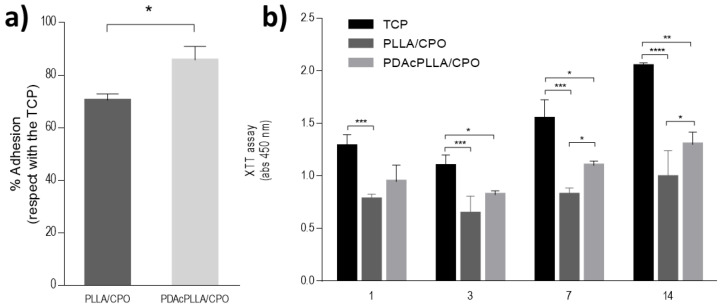
In vitro evaluation of PLLA/CPO and PDA-coated PLLA/CPO aerogels. (**a**) Cell adhesion after 24 h. Results show the percentage of adhesion with respect to the control (TCP). (**b**) Cell proliferation after 1, 3, 7, 14 days. (* *p* < 0.05; ** *p* < 0.01; *** *p* < 0.001; **** *p* < 0.0001).

**Table 1 molecules-27-02137-t001:** Specific surface area and volume of pores by N_2_ adsorption–desorption method.

Sample	Area Mesopores (m^2^/g)	Volume Mesopores (cc/g)	Area Micropores (m^2^/g)	Volume Micropores (cc/g)
PLLA/CPO	35	0.088	50	0.018
PLLA/BzOMe	81	0.25	97	0.035

**Table 2 molecules-27-02137-t002:** Settings for gels and aerogels preparation.

Sample	Dissolution Temperature	Dissolution Time	Extraction Time	Extraction Pressure	Extraction Temperature
PLLA/CPO	130 °C	240 min	180 min	2900 psi	40 °C
PLLA/BzOMe	180 °C	240 min	240 min	2900 psi	40 °C

## Data Availability

Not applicable.

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
