# Peer review of "Polydopamine-Coated Poly-Lactic Acid Aerogels as Scaffolds for Tissue Engineering Applications"

_molecules, 2022, doi:10.3390/molecules27072137_

Round 1

Reviewer 1 Report

In this manuscript, PLLa aerogels were prepared by using two different solvents. PDA coating was also applied to one type of PLLa aerogels. However PLLA aerogels and PDA coating were already reported. It is not clear what the novelty is in this study. Discussions should be improved and the importance of these different morphology and PDA coating should be better explained by using specific applications and numerical values instead of providing general information. Porosity values should also be provided. SEM images should be at the same magnification for PDA coating as well.

Reviewer 2 Report

There are acronyms used in the abstract, but some (two) are not explained: PLLA and DA.

An email address for the 1st corresponding author is probably incorrect (please double check).

The sentence in lines 111-112 is explaining α' form; it should be: "the crystalline form not including solvent molecules".

line 124: "Computational analysis showed...". There is no reference to this analysis?!

line 183: full stop not needed.

Description of Fig. 7 is misleading: there is no reference in the picture with **, but there are some with ****, but not described. See the same Fig. 9.

lines 234, 263: "Cell" from capital letter.

line 243: "etc" in the scientific article suggests that authors do not know what it stands for. It would be significant to point out all parameters referred to; see the source [41].

lines 251-252: it is definitely a description of Fig. 8, NOT 9.

line 358: is there any proof of this statement? How it was observed? It would more understandable with some graphs (just a suggestion).

References: 4, 14, 44; the year of publishing do in BOLD;

Ref. 28 does not need the authors' affiliations;

Refs: 23, 33, 39 are not complete.

Author Response

Answers to Reviewers' comments

Authors thank the referees for their comments which made it possible to better clarify work novelty aspects, to add useful information to improve results understanding and to correct oversights.

The punctual answers to the Reviewers' comments are provided below.

Referee #2:

Referee #2 comments:

  1. There are acronyms used in the abstract, but some (two) are not explained: PLLA and DA.
  2. An email address for the 1st corresponding author is probably incorrect (please double check).
  3. The sentence in lines 111-112 is explaining α' form; it should be: "the crystalline form not including solvent molecules"
  4. line 124: "Computational analysis showed...". There is no reference to this analysis?!
  5. line 183: full stop not needed.
  6. Description of Fig. 7 is misleading: there is no reference in the picture with **, but there are some with ****, but not described. See the same Fig. 9.
  7. lines 234, 263: "Cell" from capital letter.
  8. line 243: "etc" in the scientific article suggests that authors do not know what it stands for. It would be significant to point out all parameters referred to; see the source [41].
  9. lines 251-252: it is definitely a description of Fig. 8, NOT 9.
  10. References: 4, 14, 44; the year of publishing do in BOLD;
  11. 28 does not need the authors' affiliations;
  12. Refs: 23, 33, 39 are not complete.

Author's answer: all required changes have been made in the revised version of the manuscript.

Referee #2 comment: line 358: is there any proof of this statement? How it was observed? It would more understandable with some graphs (just a suggestion).

Author's answer: in line 358 of the pdf file for peer review (featured on the Molecules web site) the following statement is reported: "As the aerogel slices float in solution, only the surface at solution interface was coated". The floating of aerogel slices is an experimental data which is in perfect agreement with the low apparent density of the aerogels, which is much lower than that of the aqueous solution containing dopamine. The presence of a PDA surface coating was experimentally observed, in a sample lateral section, by the green color detected on the surface layer.

Reviewer 3 Report

Title: Polydopamine coated poly-lactic acid aerogels as scaffold for tissue engineering application

The novelty of this work, as well as its interesting results, is clearly described in the article. Therefore, the referee would like to recommend this work to minor revision and to be published after consideration according to the comments below:

  1. References should not be included in the conclusions.

  1. Please add more current papers in the literature and improve introduction section. Some interesting papers related to the topic of this manuscript could be:

Engineering with Computers, 2020, 36 (1), 359-375.

Microsystem Technologies 2018, 24 (2), 1265-1277.

Author Response

Answers to Reviewers' comments

Authors thank the referees for their comments which made it possible to better clarify work novelty aspects, to add useful information to improve results understanding and to correct oversights.

The punctual answers to the Reviewers' comments are provided below.

Referee #3

Referee #3 comment: References should not be included in the conclusions

Author's answer: references were removed from conclusions in the revised version of the manuscript.

Referee #3 comment: Please add more current papers in the literature and improve introduction section. Some interesting papers related to the topic of this manuscript could be: Engineering with Computers, 2020, 36 (1), 359-375.; Microsystem Technologies 2018, 24 (2), 1265-1277.

Author's answer: the references suggested by the referee have been added together with the following additional one: Bueno, A., Luebbert, C., Enders, S. et al. Production of polylactic acid aerogels via phase separation and supercritical CO2 drying: thermodynamic analysis of the gelation and drying process. J Mater Sci 56, 18926–18945 (2021). https://doi.org/10.1007/s10853-021-06501-0

Round 2

Reviewer 1 Report

ACCEPT